Identification and analysis of sucrose synthase gene family associated with polysaccharide biosynthesis in Dendrobium catenatum by transcriptomic analysis

Jiang Min 20110700001@fudan.edu.cn yijinsha@126.com 1 2
Li Shangyun 1
Zhao Changling 1
Zhao Mingfu 1
Xu Shaozhong 1
Wen Guosong wengs@163.com wengs@ynau.edu.cn 1
1 Research & Development Center for Heath Product, College of Agronomy and Biotechnology, Yunnan Agricultural University , Kunming , China
2 Ministry of Education Key Laboratory for Biodiversity Science and Ecological Engineering, Institute of Eco-Chongming (IEC), School of Life Sciences, Fudan University , Shanghai , China
Yin Heng
Electronic publication date: 2022 Apr 5
Publication date: 2022
Volume: 10
Electronic Location ID: e13222
Received 2021 Dec 28; Accepted 2022 Mar 14
Copyright: ©2022 Jiang et al.
Copyright year: 2022
Copyright holder: Jiang et al.
License: This is an open access article distributed under the terms of the Creative Commons Attribution License, which permits unrestricted use, distribution, reproduction and adaptation in any medium and for any purpose provided that it is properly attributed. For attribution, the original author(s), title, publication source (PeerJ) and either DOI or URL of the article must be cited.
License URL: https://creativecommons.org/licenses/by/4.0/

Keywords: SUS, Mannose, RNA-Seq, Evolution, Expression analysis, Dendrobium catenatum

Funding: National Natural Science Foundation of China 81360611 Shanghai Sailing Program 19YF1414800 This research were funded from the National Natural Science Foundation of China (81360611) and Shanghai Sailing Program (19YF1414800). The funders had no role in study design, data collection and analysis, decision to publish, or preparation of the manuscript.

==============================
Background

Dendrobium catenatum is a valuable traditional medicinal herb with high commercial value. D. catenatum stems contain abundant polysaccharides which are one of the main bioactive components. However, although some genes related to the synthesis of the polysaccharides have been reported, more key genes need to be further elucidated.

Results

In this study, the contents of polysaccharides and mannose in D. catenatum stems at four developmental stages were compared, and the stems’ transcriptomes were analyzed to explore the synthesis mechanism of the polysaccharides. Many genes involved in starch and sucrose metabolisms were identified by KEGG pathway analysis. Further analysis found that sucrose synthase (SUS; EC 2.4.1.13) gene maybe participated in the polysaccharide synthesis. Hence, we further investigated the genomic characteristics and evolution relationships of the SUS family in plants. The result suggested that the SUS gene of D. catenatum (DcSUS) had undergone the expansion characterized by tandem duplication which might be related to the enrichment of the polysaccharides in D. catenatum stems. Moreover, expression analyses of the DcSUS displayed significant divergent patterns in different tissues and could be divided into two main groups in the stems with four developmental stages.

Conclusion

In general, our results revealed that DcSUS is likely involved in the metabolic process of the stem polysaccharides, providing crucial clues for exploiting the key genes associated with the polysaccharide synthesis.

Introduction

The genus Dendrobium is a perennial medicinal herb with about 1,450 species, and widely distributes in tropical and subtropical areas such as Australia and New Guinea (Xiang et al., 2013; Zhang et al., 2016a). D. catenatum, an endangered orchid in the wild, has been served as a folk medicine nourishing “Yin”, relieving fevers and stomach upsets and enhancing immunity for hundreds of years in China (Jiang et al., 2017; Ma et al., 2018). D. catenatum is also called Dendrobium officinale under the Chinese name “Tiepishihu” and was recorded in the 2010 edition of the Chinese Pharmacopoeia (Li et al., 2017).

The fleshy stems, i.e., the main medicinal parts, of D. catenatum have abundant polysaccharides which are considered to be the main bioactive ingredients of the stems (Zhang et al., 2016b), and possess immunomodulatory, antioxidant and hepatoprotective activities (Ng et al., 2012). Therefore, the content of the polysaccharides is the main market indicator of the stem quality (Meng et al., 2013). The polysaccharides consist of mannose, glucose and arabinose, and exist in the stems with the form of 2-O-acetylglucomannan (Hua et al., 2004). Indeed, fructose and mannose are the basic building units for the polysaccharide synthesis. Cellulose synthase (CESA) gene is associated with the mannan synthesis in D. catenatum (He et al., 2015), while sucrose synthase (SUS; EC 2.4.1.13) is the main enzyme which is involved in sucrose metabolism (Koch, Wu & Xu, 1996; N’tchobo et al., 1999), and catalyzes the reversible conversion of sucrose and UDP to UDP-glucose and fructose (Nunez et al., 2008; Barratt et al., 2009). Moreover, SUS interacts with CESA as a complex to supply UDP-glucose for cell wall synthesis (Haigler et al., 2001; Ruan, Llewellyn & Furbank, 2003). The activities of sucrose invertase and sucrose-phosphate synthase (SPS) are correlated with polysaccharide levels (Wang et al., 2013), and sucrose breakdown is largely catalyzed by SUS and invertase (Huber & Huber, 1996). Although this is a general principle, rather than specific to D. catenatum, in this case, SUS and invertase should hold some correlations with the polysaccharide levels in D. catenatum. What’s more, SPS and SUS genes have been reported to be related to polysaccharide generation (Yan et al., 2015), and, in higher plants, sucrose degradation catalyzed by SUS provides the glycosyl needed in the polysaccharide synthesis (Bar-Peled & O’Neill, 2011). Hence, it is reasonable to speculate that the SUS is likely involved in the polysaccharide synthesis in D. catenatum.

SUS is encoded by a small multiple gene family that exhibits distinct, partially overlapping expression patterns and functional divergences. Identification and characterization of the SUSs in plants such as Arabidopsis thaliana (Bieniawska et al., 2007), rice (Hirose, Scofield & Terao, 2008), Populus (Zhang et al., 2011), apple (Tong et al., 2018), and pear (Abdullah et al., 2018), are helpful for understanding the physiological roles and metabolic processes of the plants. The six SUSs encoded by Arabidopsis genome are divided into three groups based on their phylogenetic relationship and genomic structures (Baud, Vaultier & Rochat, 2004; Bieniawska et al., 2007). In detail, the AtSUS1 and AtSUS2 display differential ABA-independent expressions at sugar/osmoticum levels (Dejardin, Sokolov & Kleczkowski, 1999), and the SUS1 and SUS4 are all involved in the tolerance under hypoxic conditions (Bieniawska et al., 2007). During the maturation phase, the SUS2 has highly specific expression in seeds and co-localized with the plastids in embryos (Nunez et al., 2008). Moreover, the SUS2 and SUS3 can also alter sucrose/hexose homeostasis and affect carbon partitioning and storage in developing seeds (Angeles-Nunez & Tiessen, 2010), and the SUS5 and SUS6 have C-terminal extensions relative to other isoforms and play an important role in callose synthesis in sieve plates (Barratt et al., 2009).

To date, although the key enzyme genes underlying the polysaccharide synthesis and metabolic pathway have been reported in several transcriptomes of D. catenatum (He et al., 2015; Meng et al., 2016; Zhang et al., 2016b; Shen et al., 2017), the related information of the genes remain limited. In this study, we investigated the mannose and polysaccharide contents in D. catenatum stems with four developmental stages and established the corresponding transcription databases. We also identified and characterized the SUS family involved in the polysaccharide synthesis. Eventually, the expression profiles of the SUSs in the stems were also surveyed. Our research could be helpful to further decipher the molecular mechanism of the bioactive polysaccharide biosynthesis and utilize genetic engineering to obtain abundant bioactive polysaccharides from D. catenatum.

Material and Methods

Plant materials and growth conditions

At the plantation of Yunnan Honghe Qunxin Shihu Planting Co. Ltd. of China (22°36′7″N, 103°27′36″E; Average altitude: 1,300 m; Row spacing: 12 cm × 15 cm), the stems with four developmental stages, i.e., S1 (about 2–3 months after sprouting), S2 (about 5–6 months after sprouting), S3 (about 8-9 months after sprouting) and S4 (about 11–12 months after sprouting) (Table 1; Fig. S1), were sequentially collected from the plant population of D. catenatum ‘Hongxin 6#’ with high polysaccharide content which were from the clone of a wild D. catenatum plant. The stems were divided into 2 batches. The first used for the determination of polysaccharide and mannose contents were dried at 105 °C in an oven for 10 h and then triturated by a DFT-50 pulverizer and the second, together with the roots, stems, leaves, pedicels, dorsal sepals, lateral sepals, petals, lips and columns, used for the RNA extraction were rapidly frozen in liquid nitrogen and then stored at −80°C.

Table 1 Polysaccharide and mannose content in four development stages of Dendrobium catenatum.

Sample name	Tissue description	Polysaccharide content (mg/g )	Mannose content (%w/w)	
S1	2–3 months	146	27.0	
S2	5–6 months	213	24.3	
S3	8–9 months	347	49.6	
S4	11–12 months	431	36.1	

Determination of polysaccharide and mannose contents in the stems

The polysaccharide contents in the stems with the four developmental stages detected by SGS (SGS-CSTC Standard Technical Services (Shanghai) Co., Ltd., China) were determined by using the phenol-sulfuric acid method described by the Pharmacopoeia Committee of the People’s Republic of China with glucose solutions (18, 36, 54, 72 and 90 µg/mL) as standards. 0.3 g stem powder was added with 1 mL of 5% phenol and 5 mL concentrated sulfuric acid, heated in a boiling water bath for 20 min. Finally, the absorbance at 488 nm of the reaction solution was determined with a UV-6000 spectrophotometer (Shanghai Metash, Shanghai, China). Meanwhile, the reaction solution was diluted to 2 mL with distilled water as the calibration standard. Each sample was assayed as three replicates.

For the determination of mannose contents, high performance liquid chromatography (HPLC) assays were executed with octadecylsilyl (ODS) as the filler described by the Pharmacopoeia Committee of the People’s Republic of China. 0.12 g stem powder was pre-extracted with 80% ethanol at 80°C for 4 h. Then, 100 mL double-distilled water were added to the residues and mixed with 1 mL internal standard (12 mg/mL, d-glucosamine hydrochloride, chromatographically pure, Sigma-Aldrich) at 100 °C for 1 h. Subsequently, 0.5 mL HCl (3.0 M) was mixed with the former solution and hydrolyzed at 110°C for 1 h. Then, 1-phenyl-3-methyl-5-pyrazolone (PMP) was used for derivation, incubated at 70°C for 110 min and neutralized by 0.5 mL HCl (0.3 M). The PMP labeling reaction solution was prepared by mixing with 0.3 M NaOH solution (0.4 mL) and 0.5 M PMP methanol solution (0.4 mL). To remove proteins, the sample was extracted with 2 mL chloroform for three times, mixed thoroughly for 2 min and centrifuged at 12,000 rpm for 5 min. Finally, the aqueous phase was gathered and detected by HPLC under the following conditions: ZORBAX SB-Aq C (18) column (4.6 mm × 250 mm, 5 µm); acetonitrile-0.5% ammonium acetate solution (20:80, v/v) as the mobile phase; flow rate = 1.0 mL min−1; detection wave length = 250 nm. Three biological replicates for each sample were used for the determination.

Digital gene expression library construction, sequencing and assembly

Total RNA (3 µg) from the stems were extracted by using Trizol reagent (Invitrogen, CA, USA) following the manufacturer’s recommendations, and the RNA integrity was measured by using Bioanalyzer 2100 system with RNA Nano 6000 LabChip Kit (Agilent Technologies, CA, USA) with RIN number >7.0. Sequencing libraries were constructed by using NEBNext® Ultra™ RNA Library Prep Kit for Illumina® (NEB, USA) according to manufacturer’s protocol and index codes were appended to attribute sequences to each sample. We performed only two biological replicates on each sample. The libraries were named S1-1, S1-2, S2-1, S2-2, S3-1, S3-2, S4-1, S4-2, respectively. Subsequently, the index-coded samples were clustered by cBot Cluster Generation System and then the libraries were sequenced by using Illumina Hiseq platform and 150 bp paired-end reads were generated following the manufacturer’s protocol. All raw data were cleaned by eliminating the reads containing the adapters and ploy-N and the reads with low quality, which were mapped to reference genome of D. catenatum downloaded in the Herbal Medicine Omics Database (http://herbalplant.ynau.edu.cn/) (Yan et al., 2015). Index of the reference genome was built by using Bowtie v2.2.3 (Langmead & Salzberg, 2012) and paired-end clean reads were aligned to the reference genome by using TopHat v2.0.12. We selected TopHat as the mapping tool because TopHat can generate a database of splice junctions based on the gene model annotation file and, thus, produce a better mapping result than other non-splice mapping tools. The rate of unique mapping all reached to 60% in all samples (Table 2). The square of correlation coefficients between replicates of each sample were more than 75% (Fig. S2). These raw data generated in this study had been deposited in the National Center for Biotechnology Information (NCBI) Short Read Archive (SRA) under BioProject ID PRJNA668448.

Table 2 Sequencing and assembly statistics for the 8 transcriptome data of four developmental stages in D. catenatum.

Sample ID	Raw reads (M)	Clean reads (M)	Q30 (%)	GC content (%)	No. of mapped reads (M)	Uniquely mapped reads (M)	
S1-1	63.47	62.33	95.53	46.49	44.24 (70.98%)	39.03 (62.61%)	
S1-2	65.11	64.03	95.81	46.12	46.51 (72.65%)	41.17 (64.3%)	
S2-1	76.23	74.71	95.08	46.05	52.82 (70.7%)	46.69 (62.5%)	
S2-2	61.53	60.48	95.72	46.07	43.79 (72.39%)	38.64 (63.88%)	
S3-1	63.69	62.79	95.85	45.87	45.64 (72.68%)	40.06 (63.79%)	
S3-2	72.19	71.13	95.96	45.68	50.78 (71.39%)	44.87 (63.08%)	
S4-1	64.66	63.62	96.11	45.76	43.53 (68.43%)	38.41 (60.37%)	
S4-2	75.84	74.59	96.17	45.76	53.62 (71.88%)	47.09 (63.13%)	

Differentially expressed genes (DEGs) and functional annotation

The calculation of the statistical power for our RNA-seq data was performed by RNASeqPower Calculator (Ching, Huang & Garmire, 2014). The abundance of gene expression was calculated by FPKM (Fragments Per Kilobase of transcript sequence per Million mapped reads) (Wagner, Kin & Lynch, 2012). The differential expression analysis between two randomly samples of developmental stages was executed by using the DESeq R package (1.18.0) (Anders & Huber, 2010). The DEGs were screened by EdgeR packege with fold change ≥2 and the false discovery rate (FDR) adjusted p-value <0.05 as the threshold (Robinson, McCarthy & Smyth, 2010). For gene annotation, Gene Ontology (GO) enrichment analysis of DEGs was implemented by the GOseq R package with the corrected p-values less than 0.05 as cut off (Young et al., 2010). In addition, the Kyoto Encyclopedia of Genes and Genomes (KEGG) pathway enrichment analysis of DEGs was performed by using the KOBAS software (Kanehisa et al., 2008). Likewise, the corrected p-value <0.05 was considered as statistically significant difference.

Identification of the SUS family members in plants

The 15 SUS information from the D. catenatum were downloaded from the Herbal Medicine Omics Database (http://herbalplant.ynau.edu.cn/) (Yan et al., 2015). BLASTP searches was proposed against orthologous protein sequences using Arabidopsis SUSs (Bieniawska et al., 2007) as queries in public PLAZA (http://bioinformatics.psb.ugent.be/plaza/) (Van Bel et al., 2018). Meanwhile, all retrieved gene sequences were only considered as the candidates and subjected to domain analyses by scanning in InterProscan software (De Castro et al., 2006). Importantly, the proteins only harboring sucrose synthase domain (IPR000368) rather than sucrose-phosphatase domain (IPR006380) were considered as the SUS ones. Moreover, redundancy and any alternative splice variants of sequences were eliminated. The SUSs in given species was designated according to their order on the chromosomes.

Gene structure, sequence alignment and phylogenetic analysis

All SUS protein sequences were compared with those of Arabidopsis and D. catenatum by using the Clustal Omega (http://www.ebi.ac.uk/Tools/msa/clustalo/) with default settings. The theoretical pI (isoelectric point) and Mw (molecular weight) of the SUSs were calculated by using Compute pI/Mw tool online software (http://web.expasy.org/compute_pi/). The exon/intron structures of the SUS candidates were generated by using the online Gene Structure Display Server (GSDS 2.0: http://gsds.gao-lab.org/) (Hu et al., 2015) with output in accordance with their phylogenetic tree. Phylogenetic trees of the SUSs were constructed by using MEGA 6.0 based the maximum-likelihood (ML) method with a Jones-Taylor-Thornton (JTT) model same as previous research (Tamura et al., 2013). Test of phylogeny was assessed by bootstrap method with 2000 iterations test and all positions with 95% site coverage were eliminated.

Quantitative real-time PCR (qRT-PCR) analyses

To assess the veracities of the RNA-Seq data, qRT-PCR amplification of 44 genes involved in the stem polysaccharide synthesis in four developmental stages was performed. The total RNA was extracted by using Trizol reagent and reverse-transcribed into cDNA by using PrimeScript RT Master Mix Perfect Real Time (TaKaRa) followed the manufacturer’s instructions. The qRT-PCR was executed by using SYBR®Premix Ex TaqTMII (TaKaRa) and each sample was repeated in triplicate independently. 10 µL reaction systems containing 5–50 ng of cDNA products (4 µL), 5 pmol of each primer (0.4 µL), 5 µL SYBR green master mix (2X), 0.2 µL ROX normalized fluorescent signal. The procedure for amplification was set as follows: initial activation at 95 °C for 10 min, followed by 45 cycles of 95 °C for 30 s, 60 °C for 30 s, and 72 °C for 30 s. Melting curves ranging from 60 °C to 95 °C followed by 0.5 °C/min were detected. The constitutively expression gene, D. catenatum 18S rRNA (Dendrobium_GLEAN_10067105), was used as an internal control. The primer sets were listed in Table S1. Moreover, the expression profiles of the DcSUSs in different tissues and four developmental stages were also investigated. The housekeeping gene GAPDH (NCBI accession number: KP719976) was used to normalize the relative expression levels (Jiang et al., 2017). The SUS-specific primers were shown in Table S2. The methods and conditions of the qRT-PCRs were performed as mentioned above. The relative expression levels of candidate genes were calculated by using the 2−ΔΔCt method. The up-regulated genes were defined as the ones whose fold-changes greater than 2 with the p values of <0.05, and the genes with the fold changes of 0.5 or less and the p values of <0.05 were defined as the down-regulated ones.

Results

Polysaccharide and mannose contents of the stems with four developmental stages

To provide a scientific data for taking advantage of the endangered wild Dendrobium resources, the contents of total polysaccharides and mannose, which are the most abundant neutral monosaccharide in various developmental stages of D. catenatum, were determined. D. catenatum plants had the highest polysaccharide (431 mg/g) in S4, while the highest concentration of mannose in S3 (49.6%) (Table 1). From S1 to S3, the mannose accumulated gradually, which was consistent with the variation trend of the polysaccharide. Interestingly, mannose content showed decreased in S4, rather than a sustained increase like that of the polysaccharide (Table 1), indicating that the presence of other monosaccharides had a greater impact on the polysaccharide accumulation in the later stages of D. catenatum development.

Overview of RNA-seq analysis

To elucidate the genes and metabolic pathways involved in the polysaccharide synthesis, eight transcriptomes from S1–S4 samples were studied. Due to irresistable factors, we performed only two biological replicates on each sample instead of three replicates. A total of 128.58 million, 137.76 million, 135.88 million and 140.5 million raw reads were obtained for S1, S2, S3 and S4, respectively (Table 2). Likewise, after filtering, 18.95 Gb, 20.28 Gb, 20.09 Gb and 20.73 Gb clean reads were generated, respectively (Table 2). In addition, the clean reads were mapped to D. catenatum reference genome database, and a total of 16,384 novel genes and their expression data were obtained (Tables S3 and S4). The statistical power of our RNA-seq data calculated by RNASeqPower Calculator was 0.832.

Table 3 Top 20 enriched KEGG pathways with the highest representation of DEGs among four development stages.

Pathway	Pathway ID	DEGs genes	Background number	
Starch and sucrose metabolism	ko00500	29	353	
Phenylpropanoid biosynthesis	ko00940	24	226	
Protein processing in endoplasmic reticulum	ko04141	24	355	
Circadian rhythm - plant	ko04712	24	95	
Plant hormone signal transduction	ko04075	22	327	
Plant-pathogen interaction	ko04626	19	294	
Glycine, serine and threonine metabolism	ko00260	16	126	
Phenylalanine metabolism	ko00360	16	156	
Pentose and glucuronate interconversions	ko00040	14	125	
Tyrosine metabolism	ko00350	14	69	
Glutathione metabolism	ko00480	14	153	
Isoquinoline alkaloid biosynthesis	ko00950	13	42	
Toxoplasmosis	ko05145	12	159	
Cyanoamino acid metabolism	ko00460	11	77	
Cutin, suberine and wax biosynthesis	ko00073	10	46	
MicroRNAs in cancer	ko05206	10	131	
Metabolism of xenobiotics by cytochrome P450	ko00980	9	78	
Drug metabolism-cytochrome P450	ko00982	9	100	
Ribosome	ko03010	9	444	
Phagosome	ko04145	9	139	

Go classification and KEGG analysis of the DEGs among different developmental stages

Pairwise comparisons were made to identify the candidate genes associated with the polysaccharide synthesis during development. Based on the RNA-Seq data, the distribution rang of FPKM in S1–S4 were mostly located between of 1 and 60 (Fig. S3; Table S4). The detailed information of the FPKM values was enclosed in Table S5. In addition, a total of 1762 significant DEGs were screened with the p values of ≤0.05 and the 2-fold change differences as a criteria (Fig. 1A; Fig. S4). The DEGs were shown in Table S6. To investigate the major trends between the different developmental samples in D. catenatum, these DEGs were clustered into six groups by K-means methods (Figs. 1B and 1C). Moreover, the five data sets from different comparisons were exhibited by using a Venn diagram (Fig. 1D). For example, 106 DEGs were identified in both S1 vs S3 and S1 vs S4 comparisons, while only 18 DEGs were identified in both S1 vs S3 and S3 vs S4 comparisons (Fig. 1D). To further gain insight into the functions of DEGs, the GO annotation, KEGG pathway and enrichment analyses were performed. GO annotation analysis showed that these DEGs were distributed into 70 functional terms, mainly associated with metabolic and biosynthetic process (Fig. S5 and Table S7). Meanwhile, KEGG pathway and enrichment analyses exhibited that DEGs were obviously involved in starch- and sucrose- related metabolism, phenylpropanoid biosynthesis and circadian rhythm (Table 3; Table S8).

Figure 1 Analysis of changes in gene expression among four developmental stages in D. catenatum.

(A) The number of differentially expressed genes (DEGs) was obtained from comparisons of S1 versus S2, S1 versus S3, S1 versus S4, S2 versus S3, S2 versus S4 and S3 versus S4. (B) All DEGs were classified into six clusters by short time-series expression miner (STEM, P value < 0.05). (C) Heat map illustrating the expression profiles of the developmental-differentially expressed genes. (D) Venn diagrams of the DEGs in different comparisons.

Validation and expression analysis of selected key enzyme genes related to polysaccharide biosynthesis by qRT-PCR

To verify the reliability of the RNA-seq results, we performed qRT-PCR analysis on 16 candidate genes implicated in the polysaccharide synthesis (Fig. 2). Among these, 13 genes showed high expressions with RPKM_10 in most stages, other genes displayed low expressions with RPKM_10 in all four stages. The genes selected for qRT-PCR compared with the RNA-seq data were shown (Table S1). The results showed that RNA-seq and qRT-PCR expression patterns of 16 genes were consistent, and the positive correlation coefficients (r) were all greater than 0.9 (Fig. 2). For example, the r values of Dendrobium_GLEAN_10077593 (GALE), Dendrobium_GLEAN_10128035 (GAE), and Dendrobium_GLEAN_10026326 (ASD) were 0.999, 0.999 and 0.977, respectively, indicating that the RNA-Seq data were reliable for the differential gene expression profiles in the D. catenatum stems with four developmental stages.

Figure 2 Result of qRT-PCR analysis.

The left Y axis represents RPKM value of each gene using RNA-Seq analysis. The right Y axis represents log2 transformed relative transcript amount obtained by qRT-PCR. The correlation co-efficient (r) between the two expression profiles is also showed.

Putative pathway for polysaccharide biosynthesis based on KEGG analysis in D. catenatum

Based on KEGG pathways, a total of 29 unigenes encoding 11 key enzymes involved in the polysaccharide metabolism were identified and a detailed metabolic map with the expression patterns of these genes was constructed (Fig. 3A). The detailed information of each enzyme was showed (Table S9). The largest number of unigenes (4 unigenes) were identified as beta-glucosidase (BGLU, EC:3.2.1.21) genes; the second largest number of unigenes (three) were annotated as sucrose synthase (SUS, E2.4.1.13) genes; while the third largest number of unigenes (two) were annotated as trehalose 6-phosphate phosphatase (TPP, EC:3.1.3.12) and UDP-glucuronate 4-epimerase (GAE, EC:5.1.3.6) genes. Furthermore, several unigenes were annotated as encoding UDP glucose 6-dehydrogenase (UGDH, EC:1.1.1.22), maltase-glucoamylase (MGAM, EC:3.2.1.20) and beta-fructofuranosidase (INV, EC:3.2.1.26) genes. In addition, we also showed the expression profiles of 12 genes which had differentially expressed at the four stages (Fig. 3B). Dendrobium_GLEAN_10005633 (UGDH) was highly expressed in S2 and low in S4. However, Dendrobium_GLEAN_10048133 (UGDH) was highly expressed in S3, and low in other stages (Fig. 3B), suggesting that there may be functional divergence among the members of different gene families.

Figure 3 Biosynthesis pathways of polysaccharides and related gene expression status with four developmental stages in D. catenatum.

(A) Schematic representation of the polysaccharide biosynthetic pathway based on KEGG enrichment analysis. Enzymes were highlighted and marked in orange and blue according to up-regulated or down-regulated expression in different development stages, respectively. (B) Heat map of polysaccharide biosynthetic pathway related genes in four developmental stages of D. catenatum.

Identification, evolutionary and protein domain analysis of the SUS family in D. catenatum

The polysaccharide content, transcriptome and metabolic pathway were comprehensively taken into account, we selected the SUS for further analysis. To investigate the evolution of these protein architectures across plant species, identification analyses of SUS family were conducted in 16 publicly available plant genomes. At last, 105 non-redundant SUSs were retrieved in total (Table S10). We found that the D. catenatum genome had the largest numbers of SUSs, which had subjected to prominent expansion through tandem duplication. Moreover, the Malus domestica and tetraploid soybean genomes all contained 12 SUSs, whereas a few numbers in other plant genomes were discovered, even those containing only one SUS (Table S10). Furthermore, the gene expansions might favor the generation and accumulation of polysaccharides in D. catenatum, as was similar with previous study (Yan et al., 2015). During this study, we found that the SUSs generally containing above 10 introns occupied the main points (74%), while only one SUS harbored a maximum of 23 introns (CrSUS1) (Table S10). Afterwards, the Mw and pI of the SUS proteins identified were further determined by using the online version of Compute pI/Mw tool. The Mws ranged from 10.53 (DcSUS8) to 226.02 (CrSUS1) kDa and the pIs varied from 4.72 (DcSUS8) to 9.91 (DcSUS13) (Table S10). It was noteworthy that the pIs of most SUSs (81.9%) were slightly acidic.

Then the ML tree was reconstructed based on the full-length amino-acid sequences of the 105 SUS proteins. Phylogenetic analysis showed that these SUS proteins could be divided into four families (Fig. 4). The SUS I family contained the members from green alga and seed plants, the SUS II and SUS III families gathered in angiosperm and gymnosperm, while the SUS IV family only existed in fern and moss (Fig. 4), indicating that SUS I, SUS II and SUS III families originated in common ancestral genomic contexts before the divergency of the green alga and terrestrial plants and, subsequently, non-seed and seed plant SUSs followed two distinct tracks to evolve.

Figure 4 Maximum Likelihood phylogenetic tree of sucrose synthase gene family from 16 plant species.

Phylogenetic analysis was carried with protein sequences for 105 SUS proteins from 16 plant species identified in this study.

In order to search the evolutionary relationship of the DcSUS family, a phylogenetic tree was derived from the alignments of the full-length nucleotide sequences by using the ML method by MEGA6.0 (Fig. 5A). The syntenic analyses showed that there were many duplicate pairs in the DcSUS family, including 14 dispersed duplication (DSD) and two tandem duplication (TD) ones (Table S11). The evolutionary relationship among the DcSUSs was inconsistent with that revealed in their protein phylogenetic tree. In addition, the exon/intron organization analyses of the DcSUSs were performed to examine the further genesis. Most DcSUSs had no or one intron except the DcSUS2, DcSUS4, DcSUS5 and DcSUS11 (Fig. 5B), which varied significantly from the intron numbers of other plant SUSs. The results were probably related to the incomplete annotation of the Dendrobium genome sequence. Furthermore, we further investigated the conserved domains of the DcSUSs by using InterProScan database. Generally, SUS proteins harbored a typical sucrose synthase domain (IPR000368) (Fig. 5C), but many SUS proteins also contained other ones, such as the protein kinase-like, Zinc finger and tetratricopeptide-like helical domains (Fig. 5C).

Figure 5 Phylogenetic relationships, gene and protein structure analyses in SUS proteins from D. catenatum.

(A) The phylogenetic tree was constructed from the amino acid sequences using the ML program from MEGA 6, representing relationships among 15 SUS proteins from D. catenatum. Two tandem duplicated genes were marked by blue and green, respectively. (B) The exon/intron structure of each DcSUS gene was proportionally showed based on the scale at the bottom. (c) Structure of SUS proteins in D. catenatum.

Expression profiles of the DcSUSs in different tissues and the stems with four developmental stages

Expression profiles could provide clues for their functional divergence among all members of a gene family (Whittle & Krochko, 2009). In the present research, the roots, stems, leaves, pedicels, dorsal sepals, lateral sepals, petals, lips and columns of D. catenatum were used to measure the relative expression levels of the 15 DcSUSs by qRT-PCR. All 15 DcSUSs expressed in the nine tissues. However, most DcSUS transcripts showed tissue-specific abundance patterns. For instance, DcSUS11, DcSUS13 and DcSUS14 displayed lower expressions in the pedicels, dorsal sepals and lateral sepals, and moderate expressions in other tissues. By contrast, DcSUS2, DcSUS5 and DcSUS12 expressed highly in the stems, pedicels, dorsal sepals and lateral sepals, while low expressed in other tissues (Fig. 6A). DcSUS6, DcSUS10 and DcSUS15 had low expressions only in the stems, while high expressions in other tissues. DcSUS3 and DcSUS4 showed moderate expressions in the pedicels, dorsal sepals and lateral sepals, but high expressions in other tissues. It was interesting that the expression pattern of the DcSUS8 was just opposite. In addition, the DcSUS1 and DcSUS7 displayed a relatively lower expressions in all tissues, while the DcSUS9 displayed relatively high expressions in all tissues (Fig. 6A). Therefore, the DcSUSs might exhibit function divergences in the corresponding tissues.

Figure 6 Expression profiles of D. catenatum DcSUS genes in different tissues and developmental stages.

(A) Expression patterns of DcSUS genes in nine tissues including root, stem, leaf, pedicel, dorsal sepal, lateral sepal, petal, lip and column. (B) Expression patterns of DcSUS genes in four developmental stages of D. catenatum.

To survey the development-dependent expression difference of the DcSUSs, we also investigated their expression patterns in the stems with four developmental stages (Table S12). Their expression profiles could be mainly divided into two cases. One had low expressions in four developmental stages including the DcSUS1, DcSUS5, DcSUS6, DcSUS10 and DcSUS15. Another displayed relative high expression including DcSUS2, DcSUS3, DcSUS7, DcSUS8, DcSUS9, DcSUS12, DcSUS13 and DcSUS14 (Fig. 6B). Moreover, the DcSUS11 showed high expression only in S1 stage, and low expressions in other three stages. While the DcSUS4 had a moderate expression only in S1 stage, and high expressions in other stages (Fig. 6B). Thus, the DcSUSs might hold distinct functions in different development stages of the stems.

Discussion

Potential candidate genes involved in polysaccharide synthesis

The dominating bioactive components of D. catenatum were soluble polysaccharides (Ng et al., 2012), which were synthesized from monosaccharides, such as glucose, mannose, galactose, rhamnose, arabinose, xylose, and so on (Zha et al., 2007; Fan et al., 2009). It was indicated in the current research that, the longer the plants grow, the higher the content of soluble polysaccharides (Table 1). In addition, the transcriptomes of the stems with the four developmental stages revealed 1762 DEGs (Fig. 1). Moreover, further analysis of KEGG pathway displayed that many genes encoding the key enzymes involved in starch and sucrose metabolism were identified. The results showed that beta-fructofuranosidase (INV, EC:3.2.1.26) expressed differentially in the developmental process of D. catenatum stems (Fig. 3A). A previous study has reported that the elongation of A. thaliana roots was regulated by vacuolar invertase (INV) that can degrade sucrose to produce glucose and fructose (Sergeeva et al., 2006). Moreover, the SUS (EC 2.4.1.13) gene associated with sucrose metabolism had significant differentially expressed among developmental stages (Fig. 3A). Moreover, Os4bglu12 β-glucosidase (EC:3.2.1.21) had high exoglucanase activity and consistent with a role in cell wall metabolism (Opassiri et al., 2006), which similarly had distinct expression profiles in our study (Fig. 3A). Likewise, a previous research reported that UDP-arabinose 4-epimerase (UXE, EC:5.1.3.5) with distinct expression patterns (Fig. 3) played key roles in the synthesis of arabinosylated cell wall components (Burget et al., 2003). Furthermore, UDP-glucose dehydrogenase (UGDH, EC 1.1.1.22) catalyzes the formation of UDP-GlcA from UDPGlc, and then synthesizes UDP-Xyl (Oka & Jigami, 2006) in which d-xylose is mainly present in the form of cell wall polysaccharides and N-glycan (Mercx et al., 2017). However, there were also some genes that had no obvious differentially expressions in the present study, but they had been previously proved to be involved in polysaccharide synthesis. For example, mannose-1-phosphate_guanylyltransferase (EC:2.7.7.13) was a key enzyme associated with the cellulose biosynthesis (Kim et al., 2014), which had been identified to be involved in the synthesis of mannan polysaccharides in D. officinale (D. catenatum) (He et al., 2015). In addition, mannose-6-phosphate isomerase (MPI, EC:5.3.1.8) catalyzed the reversible isomerization between D-fructose 6-phosphate and D-mannose 6-phosphate and participated in hexose metabolic process (Wang et al., 2014). Interestingly, MPI encoding genes had highly expression levels in the D. catenatum stems with four developmental stages (Fig. 3B). In a word, these enzymes might contribute to the growth and development of D. catenatum and their genes could be used as the candidate ones involved in the synthesis of the polysaccharides.

Tandem duplication contributed to DcSUS expansion

Tandem duplications generating the duplicates that are closely adjacent to each other (generally separated by 10 or fewer genes) has facilitated greatly to the expansion of plant gene families (Rizzon, Ponger & Gaut, 2006; Freeling, 2009). Our analysis indicated that there were obvious gene expansions in the DcSUSs holding the most gene family members (Table S11). Further syntenic analyses displayed that the multiple copies of the DcSUS resulted from tandem duplications or segmental duplications (Fig. 5A; Table S11), which was similar to those of MKK family (Jiang & Chu, 2018; Jiang, Li & Wang, 2021). Moreover, previous studies also reported that the DcSUSs maybe underwent expansion through tandem duplication (Yan et al., 2015; Qiao et al., 2019). What’s more, these tandem duplicates generally played a vital role in the plant adaptation to respond rapidly changing environments (Hanada et al., 2008). Similarly, the expansions of SPS and SUS (SuSy) had been believed to contribute to the polysaccharide richness in D. officinale (i.e., D. catenatum) (Yan et al., 2015). Meanwhile, just as the synthesis of the polysaccharide in A. thaliana was beneficial to the drought-resisting of A. thaliana (Balsamo et al., 2015), the polysaccharide synthesis and the expressions of the related genes including the DcSUS in D. catenatum stems contributed to the adaptability of cultivated D. catenatum to various stresses, e.g., drought, salt and osmotic ones (Huang et al., 2021). As a result, since the genes giving plants the ability to respond rapidly changing environments have been verified to hold expansion tendency including tandem duplication (Hanada et al., 2008), the richness of polysaccharides in D. catenatum might be related to the expansion of these genes including the DcSUS, which was in accordance with that reported in previous studies (Yan et al., 2015).

Sequence evolution, domain organization and expression patterns of DcSUS

Compared with other species, there were more large SUS numbers in Glycine max, M. domestica and D. catenatum genome (Table S10). Generally, the increase in genes is associated with polyploidy or ancient polyploidization events (Jiao et al., 2011). For example, the 12 SUSs existed in recently duplicated tetraploid G. max genomes which had occurred two rounds of whole genome duplications (WGDs). In addition, except the sucrose synthase domain, other domain organizations were detected, e.g., the protein kinase-like and tetratricopeptide-like helical domains (Fig. 5C). The glycosyltransferase (GT) gene family affect many aspects of plant growth and development. For instance, the loss of GT genes affect the production of recombinant proteins beta-1, 2-xylose and core alpha-1,3-fucose in Nicotiana benthamiana (Jansing et al., 2019), and the UDP-glycosyltransferase gene regulates the ginsenoside synthesis in Panax ginseng and Panax quinquefolius (Lu et al., 2017). Moreover, Magnaporthe oryzae Chitinase MoChia1 interacts with OsTPR that is a tetratricopeptide repeat protein to counteract the function of this fungal chitinase and regain immunity (Yang et al., 2019). These results indicated that the SUSs might also have divergent functions due to the domain organization change.

In a number of cases, the changes in tissue-specific expression of genes may also lead to the changes in the functions of the gene paralogs (Zhang & Ma, 2012). VvSUS3 was the most highly expressed gene in the berries, which closely affected the sugar content in the berries and the changes in SUS activity (Ren et al., 2020). The carrot DcSus were higher expressed in the leaf blades than those in the roots and petioles, which showed strong negative correlation both with the sucrose and soluble sugar contents (Liu et al., 2018). In addition, most MdSUSs displayed decreased expressions during fruit development, whereas the expression profiles of MdSUS2.1 and MdSUS1.4 were opposite, which indicated that MdSUSs might play distinct functions in the sugar utilization and sink-source sugar cycle in apple (Tong et al., 2018). Therefore, we investigated the expression profiles of DcSUSs in different D. catenatum tissues. Most DcSUSs displayed tissue-specific abundance patterns, especially in the stems (Fig. 6A), implying that, in sucrose metabolism, the DcSUSs might play distinct roles different from those in other plants. For example, DcSUS7 showed higher expression in stems than those in other tissues (Fig. 6A), indicating that the gene was most likely related to the synthesis of the sucrose in D. catenatum stems. Furthermore, DcSUS10 had higher expressions in S3 and S4 stages than that in S1 (Fig. 6B), which was similar to the polysaccharide and mannose contents in the stems with the specific stages (Table 1), suggesting that the sucrose synthesis in developing stems might be closely associated with the DcSUSs.

Conclusion

In this study, we detected and compared the polysaccharide and mannose contents of the D. catenatum stems with four developmental stages, and conducted RNA-seq analysis of the corresponding stems. A total of 16,384 genes were detected. Several DEGs correlated with the metabolic and biosynthetic process were identified. Further analysis showed that the DEGs were mainly enriched in starch and sucrose metabolism pathway. More importantly, we observed the SUS encoding polysaccharide biosynthase in 16 representative plants, and studied its genomic characteristics and evolutionary relationships. The results suggested that the expansions in DcSUSs were caused by tandem duplications. Moreover, the 15 SUSs showed two different expression patterns at the four developmental stages and were significantly regulated in different D. catenatum tissues. In general, these results not only provided gene resources for the genetic improvement of D. catenatum, but also laid a foundation for further understanding of the molecular mechanism of polysaccharide biosynthesis.

Supplemental Information

Figure S1 The morphological characteristics of four development stages of D. catenatum.

Click here for additional data file.

Figure S2 Correlation coefficients between RNAseq biological replicates

Click here for additional data file.

Figure S3 The FPKM distribution of four development stages

Click here for additional data file.

Figure S4 The Volcano Plot of differentially expressed genes (DEGs) was obtained from comparisons of S1 versus S2, S1 versus S3, S1 versus S4, S2 versus S3, S2 versus S4 and S3 versus S4

Click here for additional data file.

Figure S5 The top significant GO terms and pathways of the DEGs

The results are summarized in mainly three categories: biological process, cellular component and molecular function.

Click here for additional data file.

Table S1 The list of qRT-PCR primers of genes selected in D. catenatum

Click here for additional data file.

Table S2 The list of qRT-PCR primers of SUS genes in D. catenatum

Click here for additional data file.

Table S3 The list of novel genes in compared to previously genome assembly in our RNA-seq analyses

Click here for additional data file.

Table S4 The statistic results of FPKM interval about 8 samples in the study

Click here for additional data file.

Table S5 Expression patterns of all genes in the 8 samples

Click here for additional data file.

Table S6 The list of differentially expressed genes (DEGs) was obtained from comparisons of S1 versus S2, S1 versus S3, S1 versus S4, S2 versus S3, S2 versus S4 and S3 versus S4

Click here for additional data file.

Table S7 The GO enrichment of DEGs in six different comparisons

Click here for additional data file.

Table S8 The KEGG enrichment analysis of DEGs in six different comparisons

Click here for additional data file.

Table S9 The list of full names about enzymes by EC IDs as mentioned in Figure 3

Click here for additional data file.

Table S10 The lists of SUS gene family in 16 plant species

Click here for additional data file.

Table S11 The duplicated genes of SUS gene family in D. catenatum chromosomes

Click here for additional data file.

Table S12 The raw data of qRT-PCR result

Click here for additional data file.

Additional Information and Declarations

Competing Interests

Author Contributions

Data Availability

The authors declare there are no competing interests.

Min Jiang conceived and designed the experiments, performed the experiments, analyzed the data, prepared figures and/or tables, authored or reviewed drafts of the paper, and approved the final draft.

Shangyun Li performed the experiments, analyzed the data, prepared figures and/or tables, and approved the final draft.

Changling Zhao analyzed the data, authored or reviewed drafts of the paper, and approved the final draft.

Mingfu Zhao and Shaozhong Xu analyzed the data, prepared figures and/or tables, and approved the final draft.

Guosong Wen conceived and designed the experiments, analyzed the data, prepared figures and/or tables, authored or reviewed drafts of the paper, and approved the final draft.

The following information was supplied regarding data availability:

The sequences are available at NCBI’s Short Read Archive (SRA): PRJNA668448.

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
