# Peer review of "Identification and analysis of sucrose synthase gene family associated with polysaccharide biosynthesis in Dendrobium catenatum by transcriptomic analysis"

_PeerJ, doi:10.7717/peerj.13222_

## Round 0.1 · original submission · Major Revisions

The reviewers and I think this paper is helpful in understanding the molecular mechanism of polysaccharide metabolism. However, there are many issues that need to be revised.

Reviewer 1 ·

Basic reporting

In this work, the authors comprehensively investigated sucrose synthase genes in the plant species Dendrobium catenatum. With the implementation of metabolic, transcriptomic, KEGG enrichment, and phylogenetic analyses et al., the authors revealed expression patterns of SUS genes in various tissue types at different developmental stages in relation to relevant polysaccharide accumulation, as well as their evolutionary trajectories. This work helps understand the molecular mechanism of polysaccharides metabolism. I recommend publication of this manuscript after minor revision.

Experimental design

Experiments are overall well designed and clearly described, the authors may want to include the information regarding how the transcriptomes were assembled.

Validity of the findings

Findings and conclusions are generally well supported, I don’t have major comments, certain typos need to be corrected, e.g., line 149, line 213, fig. 6a.

Additional comments

No

·

Basic reporting

In this study, Jian et al. used the four different developmental stem stages of Dendrobium cantenatum to compare the contents of polysaccharides and mannose and analyze their transcriptomes to explore biological mechanism of polysaccharides biosynthesis. This study is different from other previous related studies. Thus, their results could be formally published. However, at present the manuscript has some apparent weak points as showed below:
1) The authors simply described the sampling method as that the four developmental stages of D. catenatum stems of ‘Hongxin 6#’ plants, were sequentially collected from the plantation (Honghe, Yunnan, China) for paired-end transcriptome sequencing. Generally, the catenatum dendrobums farms get the seedlings from the seeds and different seedlings have diversity genotypes. More important, several catenatum dendrobums individuals are grew together to form a cluster in the farms. Thus, it is possible that different stems from same cluster have various genotypes. Therefore, it is necessary for authors to check their sampling materials in the four developmental stages of D. catenatum stems are the same genotype or different genotypes.
2) If different stems from the same cluster have different genotypes, the authors need to be careful about their conclusions, such as lines 467-470, “Furthermore, DcSUS10 had higher expression in S3 and S4 than that in S1 (Fig. 6b), similar to their polysaccharides and mannose content (Table 1), suggesting that the sucrose synthesis in different stem development might be closely involved with DcSUS genes”. It is possible that the results of DcSUS10 having higher expression in S3 and S4 than that in S1 (Fig. 6b), similar to their polysaccharides and mannose content are caused by different genotypes rather than the different development stages.
3) Lines 68-73: The authors stated that “The activities of sucrose invertase and sucrose-phosphate synthase (SPS) are correlated with polysaccharide levels (Wang et al., 2013), and sucrose breakdown is largely catalyzed by SUS and invertase (Huber and Huber, 1996). SPS and SUS genes have been reported to be related to polysaccharide generation (Yan et al., 2015). Hence, it is reasonable to speculate that SUS gene is likely also involved in the polysaccharide biosynthesis in D. catenatum”. In fact, Wang et al. (2013) suggested that the activities of sucrose invertase and sucrose-phosphate synthase (SPS) are correlated with polysaccharide levels in D. catenatum. However, the conclusion that sucrose breakdown is largely catalyzed by SUS and invertase by Huber and Huber (1996) is a general principle rather than especially for D. catenatum. It is not clear that this general principle can be used in D. catenatum or not. Moreover, Yan et al. (2015) only listed the potential genes that related with polysaccharide generation based on their whole genome sequence results. Therefore, it is not clear how the authors have got the speculation that SUS gene is likely also involved in the polysaccharide biosynthesis in D. catenatum.
4) Lines 430-433: the authors said that “These tandem duplicates generally played vital role in plant adaptation to response rapidly changing environments (Hanada et al., 2008). Therefore, we have reason to speculate the richness of polysaccharides in D. catenatum might be related to the expansion of these genes”. The authors have not given out any information about the D. catenatum has adapted to response rapidly changing environments. How the authors can speculate that the richness of polysaccharides in D. catenatum might be related to the expansion of these genes?
5) Line 412: The authors used one plant name D. officinale. Actually, both the names of Dendrobium catenatum and D. officinale are the same entity. So, this sentence can be changed as “which had identified involved in biosynthesis of mannan polysaccharides in D. officinale (D. catenatum) (He et al., 2015)”.

Experimental design

Fine

Validity of the findings

Nice

Additional comments

No

Reviewer 3 ·

Basic reporting

In the manuscript entitled “Identification and analysis of sucrose synthase gene family associated with polysaccharide biosynthesis in Dendrobium catenatum by transcriptomic analysis”, the authors used bioinformatic methods to analyze the sucrose synthase gene in Dendrobium catenatum. The amount of data analyzed is large and the results obtained are abundant.

I believe that this article can meet the requirements published in Peerj after reversion.

Comments
1. The organization of paper writing also needs to be further improved.
2. Overall, major text in the discussion looks fine. However, a better discussion will help improve the manuscript.
3. Reference needs further examination. Retain the relevant and recent literature/report.
4. Before being considered for publication, the manuscript will be corrected by a fluent English speaker, alternatively please use one of the commercial English language editing services available.

Experimental design

well

Validity of the findings

well

---

## Round 0.2 · accepted · Accept

Congratulations. Both reviewers think that you have answered their questions properly, so we think this paper is acceptable for publication. However, when revising the proof, maybe you can add some information about the variety of 'Hongxin 6#'.

Reviewer 1 ·

Basic reporting

The authors have carefully addressed my comments, I suggest accepting in its current form.

Experimental design

Well

Validity of the findings

Well

Additional comments

No

·

Basic reporting

The authors have addressed all questions raised by reviewers. I have no more questions. However, I think it will be better if the authors can give more infromation about the variety of ‘Hongxin 6#’.

Experimental design

No comment

Validity of the findings

no commment

Additional comments

no